# Hallway Gait Monitoring System Using an In-Package Integrated Dielectric Lens Paired with a mm-Wave Radar

**DOI:** 10.3390/s23010071

**Published:** 2022-12-21

**Authors:** Hajar Abedi, Jennifer Boger, Plinio Pelegrini Morita, Alexander Wong, George Shaker

**Affiliations:** 1Department of Systems Design Engineering, University of Waterloo, Waterloo, ON N2L 3G1, Canada; 2School of Public Health Sciences, University of Waterloo, Waterloo, ON N2L 3G1, Canada; 3Electrical and Computer Engineering Department, University of Waterloo, Waterloo, ON N2L 3G1, Canada

**Keywords:** gait extraction, FMCW radar, dielectric lens antenna, spatiotemporal gait parameters

## Abstract

This paper presents a novel hallway gait extraction system that enables an individual’s spatiotemporal gait parameter extraction at each gait cycle using a single FMCW (Frequency Modulated Continuous Wave) radar. The purpose of the proposed system is to detect changes in gait that may be the signs of changes in mobility, cognition, and frailty, particularly for older adults in retirement homes. We believe that one of the straightforward applications for gait monitoring using radars is in corridors and hallways, which are commonly available in most retirement and long-term care homes. To achieve in-corridor coverage, we designed an in-package hyperbola-based lens antenna integrated with a radar module package empowered by our fast and easy-to-implement gait extraction method. We validated system functionality by capturing spatiotemporal gait values (e.g., speed, step points, step time, step length, and step count) of people walking in a hallway. The results achieved in this work pave the way to explore the use of stand-alone radar-based sensors in long hallways in retirement apartment buildings or individual’s homes for use in day-to-day long-term monitoring of gait parameters of older adults.

## 1. Introduction

Human gait and activity monitoring play important roles in many applications that can significantly improve the ability to be as independent, secure, and healthy as possible [1]. Notably, gait qualities are increasingly being recognized as a measure of a person’s health status; changes in normal walking patterns, such as gait speed reduction or lack of balance, can signify a change in cognition and an increase in the probability of a fall occurring, especially for older adults [2]. However, gait speed variations due to cognitive or other conditions may go undetected as the effect is gradual and often not noticeable during clinic visits. Technologies that can detect changes in people’s gait patterns could be used to support the detection, evaluation, and monitoring of parameters related to changes in mobility, cognition, and frailty [3]. Early detection of such changes results in a higher probability they can be better supported, which in turn can increase the probability of independence and quality of life of the person being monitored [3]. This approach is especially relevant to the growing population of older adults, most of whom wish to remain in their own homes. 

Current gait monitoring systems such as the GaitRite mat [4] and Vicon [5] can make precise measures of gait, however such systems are expensive and difficult to operate, making them impractical for day-to-day gait monitoring. Infrequent measurements may not detect gradual changes, thus delaying diagnosis. On the other hand, researchers and technology companies have associated and targeted falls with adverse events [6]. Falling is a leading cause of mortality and major injuries among the elderly (each year, 3 million older people are treated in emergency departments for fall injuries) [6]. To minimize long-term debilitation and maintain independence and quality of life, generating day-to-day data on gait qualities is of paramount importance (especially for older adults). Daily gait assessment would lead to monitoring possible deterioration in health and preventing future health problems.

Cameras are more affordable and less restrictive but are prone to difficulties with different lighting conditions as well as privacy concerns. Wearables can be used for frequent measurements but require people using them to remember to wear them and remember to charge. Therefore, there is a pressing need for an affordable, easy-to-use technology that can measure human gait parameters continuously, unobtrusively, and reliably if we are to get a better understanding of people’s true gait and how their gait may change over time. 

The use of a radar system to monitor gait is appealing due to its reliable functionality during different lighting conditions, the avoidance of having to wear, carry, or charge a device, and protection of privacy [7,8,9,10,11]. CW (Continuous Wave) radars have been explored through various studies to extract gait characteristics [12,13,14,15,16,17,18,19], however, an extra device (e.g., a treadmill) is required to obtain both spatial and temporal parameters because the position of the subject could not be obtained by a CW radar. For example, in [19], two CW radars and a treadmill were used to extract spatiotemporal parameters. Apparently, the extra devices increase cost while adding too much complexity, thus not suitable for frequent in-home/hallway gait assessment [19]. Only one FMCW radar could be used to extract gait spatiotemporal parameters because it provides the range and Doppler signature simultaneously.

Another challenge in gait assessment is that human walking contains the micro-Doppler signature that is dependent on the direction of the motion. To overcome the dependency on the relative angle between the radar and a walking person, we decided to monitor human gait in a corridor or hallway, as this is something that is in virtually every place people live and results in people performing a relatively straight line of walking in a natural way several times a day. However, as will be shown in this work, walls and other objects in the hallway have a strong “clutter” impact, creating multipath reflections when wide beam radar antennas are used. The multipath reflections could result in an inaccurate gait measurement because gait extraction algorithms employ the assumption that the maximum reflected signals come from the torso of the walking person (rather than indirect reflections or multipath) [19]. For instance, reflected signals from the torso in the spectrogram were obtained in [19] to extract stride rate based on the assumption that the maximum signals are the torso’s returns. However, this assumption is correct only if the person was walking in a large clutter-free environment. 

To eliminate the multipath reflections, we propose two viable solutions: 1. novel signal processing and 2. radar antenna modification. In our previous work [11], we put forward a novel radar signal processing and unsupervised learning algorithm to track a person walking in the hallway and distinguish the direct signals from multipath reflections. The results provided in [11] showed that the proposed association and tracking method could reliably track the walking person and extract spatiotemporal parameters accurately. However, multiple transmitters and receivers are required to obtain a range-azimuth heatmap of the environment for single-person monitoring. Additionally, since the proposed method is based on temporal clustering and association, it requires a powerful system due to the computationally complex and time-consuming processes. Thus, our signal processing methodology can be improved; it shows promise and is being refined; it will be the topic of future publications. 

The purpose of this work is to showcase the effectiveness of radar antenna modification in mitigating multipath reflections in hallway gait monitoring without the need for complex signal processing methods. We have been using commercially available low-cost radars that typically have wide beamwidth and suffer from low gain and short-range coverage [11,20,21,22,23,24]. In [23], we proposed a hyperbolic 3D printed dielectric lens to sharpen the radiation beam of an FMCW radar antenna to mitigate multipath reflections. In this work, we integrated the dielectric lens proposed in [23] within the radar package, thus, creating a combined module with a narrower beamwidth. This in-package “covert” lens antenna with a sharper beam enables both the radar and processing technique to focus on processing the radial velocity, which in turn means that we can deploy Doppler/micro-Doppler processing for obtaining gait characteristics more efficiently. In addition to the lens integration, we demonstrate that with a proper combination of an in-package lens antenna and a faster and easy-to-implement signal processing technique, our proposed system is capable of reliably extracting spatiotemporal parameters at each gait cycle for an individual. Furthermore, since our modified system is empowered by a relatively fast and easy-to-implement algorithm that uses only one transmitter and one receiver, the processing chain could be implemented in a radar DSP or a Raspberry Pi [21,25] without a need for an extra powerful desktop or a laptop system. 

In this paper, we achieved a reliable hallway gait assessment system using a commercially available radar without altering the radar PCB board while enabling a simple flat packaging for the whole system. Our work provided a useful and valuable approach for many radar-based applications that do not desire the use of hemispherical lens designs. The system we outline below is advantageous in that it is a stand-alone, portable, and easy-to-use device that can measure spatiotemporal gait parameters accurately and reliably in a highly cluttered environment simply and cost-effectively compared to other current alternatives. Therefore, our main contributions to this paper are as follows: A reliable stand-alone hallway gait assessment system, which is of great significance for building an affordable everyday gait monitoring system.An innovative method, including the choice of the package-friendly lens and inclusion as part of the package design to be paired with a commercially available radar that could remove/mitigate multipath signals and extract gait parameters in such a cluttered environment in the hallway.Implementing a fast and easy-to-implement gait extraction algorithm to extract spatiotemporal gait parameters at each single gait cycle, such as speed, step points, step length, stride length and step count, using only one FMCW radar sensor.

## 2. Hallway Gait Monitoring System

We have been using a commercially available mm-wave FMCW radar system [11,20,21,22,23,24,26] (AWR1443Boost) operating at the frequency range of 77–81 GHz. One challenge we encountered was the wide beam of the radar antenna covering a wide range of the surrounding environment creating multipath effects [23]. Hence, high directive antennas are required to overcome the aforementioned issue in long hallway gait monitoring. Low-cost 3D printing could be used to develop a custom-built lens that can be paired with the existing radar antenna design to achieve a narrow beamwidth [23]. 

### 2.1. Lens Design for Hallway Gait Monitoring System

The dielectric lens design is commonly based on the assumption that the overall lens dimensions and the surface radius of the curvature at any point are relatively larger than the wavelength [27]. This assumption leads to neglecting the diffraction of Electromagnetic waves passing through the edges of the lens. To collimate waves from a feeder (assuming a point source), all rays at the outer surface of the lens should have the same phase. Hence, the electrical path length of every ray needs to be the same at the exiting wavefront to transform all waves to plane waves [27]. To make the system ideal for long hallway gait monitoring, owing to ease of fabrication and aesthetic packaging (since it presents an outer planar surface) [27], an in-package hyperbola-based dielectric lens was designed and integrated with the FMCW radar. Figure 1a shows the schematic of the designed system. In a hyperbolic lens, the refraction occurs in the hyperbolic surface where the waves enter. In this configuration, the other side where waves exit is planer and does not refract the rays. The shape of the outer surface of the hyperbolic lens in the polar coordinates can be obtained by imposing the path length collimation condition and the actual physical length condition. Hence, the thickness of the lens (*t*) in terms of *F*, *r*, and *n*, can be written as [27]:(1)$$t\left(r\right)=\frac{F}{n+1}\left(\sqrt{1+\frac{n+1}{n-1}{\left(\frac{r}{F}\right)}^{2}}-1\right)$$
where *F* is the focal length of the hyperbolic or the distance to the feed, *n = √ε_r_*, 0 < *r* < *R*, *R* is the radius of the hyperbola, and *ε_r_* is the relative permittivity of the lens. Therefore, a proper design of the lens is based on (1). Rays coming from the primary feeds are collimated, and the antenna system gain will be improved. In this work, we base our design of the in-package lens on our previously used radar sensor [11], although the same design procedure should be applicable to other types of radar sensors. We point out that while one transmitter and one receiver are enough for the single-person gait monitoring proposed in this work, we designed the in-package lens for the same Multiple Input Multiple Output (MIMO) radar used in our previous work [11]. There were two reasons for this: 1. To provide a reliable and reasonable comparison (in terms of operation frequency, range resolution, etc.) between the outcome of this work and our previous work presented in [11], 2. To use the same fabricated system for multiple people monitoring in our future work. Note that for multiple people monitoring, the information of all transmitters and receivers should be used, as demonstrated by our work in [28]. 

As shown in Figure 1a, since the lens should be paired with the radar antennas, the radar antennas are primary feeds for the hyperbolic lens. We used polylactic acid (PLA) as the building material for our design because of its low-cost, low loss, and ease of fabrication. To initiate the simulation and design process, we first performed 3D printed material characterization to find the electrical properties of the printed material using the open-ended coaxial probe DAK1.2E-TL [29]. This was done to check and correct for potential fabrication errors as well as possible instability of the dielectric constant of 3D printed material versus filling ratio and varying temperature. The measured parameters (*σ* = 0.19 s/m and *ε_r_* = 2.68) were used to find the optimum design parameters of the hyperbolic lens antenna in the High-Frequency Structure Simulator (HFSS) [30]. To find the optimum lens parameters, comprehensive parametric analyses of the effect of the lens dimension and the focal length on the radiation pattern were intensively investigated. Based on the simulation results from parametric optimization analyses, a hyperbola-based lens with a value of *R* = *F* = 10 λ (λ: wavelength) was selected; Figure 1b shows the entire system. To explore the effectiveness of the hyperbola-based lens antenna in beam sharpening and gain improvement, simulated and measured patterns (H-plane) of the radar received power transmitted by the TX_1_ and received by the RX_1_ integrated with and without the lens is shown in Figure 2. For this work, we provided the results obtained from TX_1_ and RX_1_; other receivers yielded similar results. The results show good agreement between simulation and measured results, with the radar integrated with the dielectric lens achieving a significantly sharpened beam of the radar antenna. Moreover, measured results of the intensity of the received power of signals of the radar antenna show that the received power from the radar integrated with the lens is sharper and more intensive (i.e., the lens provides more than 14 dB improvement in gain). To evaluate the system for our specific application, we require a gait extraction algorithm to evaluate our proposed system and assess human gait in the hallway using our proposed hardware design, which we present in the following subsection.

### 2.2. Gait Extraction Algorithm

For each walking cycle, each segment of the human body creates different micro-Doppler effects that can be observed in the spectrogram [10]. It has been shown that since the torso constitutes a significant part of the reflected signals, the torso line can be selected by isolating the maximum signal [10]. However, this assumption is particularly valid when gait is extracted in a large clutter-free environment; stationary objects in a cluttered space create multipath or ghosting effects that can invalidate this assumption. The amplitude of the multipath reflections could be more than the subject’s torso line if the phases of the reflected signals are constructive [11]. Therefore, the maximum value of the reflected signals might not represent the walking subject’s position over time. However, in this work, we demonstrate that since the sharp antenna beam mitigates the multipath reflections significantly, the assumption holds true even in a cluttered environment such as a hallway. For instance, Figure 3a shows the range of the walking subject in a hallway using the radar without the lens. Frame 150 is used to illustrate the discussion below; the same results could be obtained from the other frames. 

As shown in Figure 3, the amplitude of the multipath reflection is more than the amplitude of the subject’s position (the green circle). Additionally, various multipath reflections can clearly be seen at various range bins (shown by the red circle). However, Figure 3b shows that the lens mitigates multipath reflections effectively. As a result, the range bin of the subject’s torso position (the green circle) corresponds to the maximum amplitude representing the subject’s position. Hence, instead of implementing a potentially time-consuming and computationally expensive signal processing technique to remove the multipath effects, using the proposed in-package system and isolating the torso’s range bin, we could obtain a reliable hallway gait extraction system. Therefore, the main difference between this work and our previous work is that in [11] the Capon beamformer, 2D-CFAR (constant false alarm rate), DBSCAN (density-based spatial clustering of applications with noise) clustering and an association and tracking algorithm were proposed to obtain the position of a walking subject over time and remove the ghosting effects. However, in this work, the position of a walking subject can be approximated by taking the maximum value in the range bins. This is done due to the proper radar setup and antenna radiation pattern modification. Therefore, only one transmitter and one receiver are sufficient for single-person hallway gait monitoring using our in-package lens without the need to generate the 2-D heatmap of the environment.

The block diagram of our proposed gait extraction algorithm is depicted in Figure 4. In this paper, after obtaining the range bin of the subject, we follow similar processes presented in our previous work [11] to extract gait values. Having the range bin of the walking subject over time, the speed and other spatiotemporal parameters could be calculated; we refer interested readers to [11] for more details on these topics. Since our focus in this paper is on showing the performance of our algorithm integrated with the in-package lens, gait parameters of a single subject were extracted. Therefore, only one transmitter and one receiver are sufficient to provide range and micro-Doppler information on a subject. It should be pointed out that adding more transmitters and receivers would not solve the issue of multipath effects or ghosts in our gait monitoring application. Even using the MIMO channels, as shown in [11], we still require a sophisticated method to mitigate the multipath effects to obtain accurate gait parameters [11]. As mentioned, the main difference between this work with our previous work is that instead of using several sophisticated and time-consuming algorithms to remove multipath effects, we demonstrate that a single transmitter and receiver antenna with a thin beamwidth can provide accurate gait parameters in a cluttered-environment such as a hallway.

## 3. Experimental Results

In this section, we provided the results of gait assessments in two different environments: a hallway and a large environment without clutters. 

For all experiments conducted in a hallway environment, participants were asked to walk 420 cm toward and away from the radar three times. According to the walking distance, the radar configuration parameters used for this experiment are defined and listed in Table 1; radar parameters, definition and values used for radar configuration are to be used for the tests. 

As explained in Section 2.2, our gait monitoring extraction algorithm is based on the fluctuation of the torso’s speed during the stance and swing phase. Thus, it is crucial to obtain the torso’s range bin accurately. To associate the maximum of the reflected signals to a walking subject’s torso, the radar should be located at a position where its main beam illuminates the torso, otherwise, a range bin corresponding to legs or arms could be obtained. From the Lund & Browder chart [31], the torso contributes approximately 13% of the total body surface area, which is more than the surface area of other parts. Therefore, with an appropriate radar setup, an individual’s torso constitutes a major part of the reflected signals. There are two key factors in finding the best position to place the radar sensor: 1. Radar antenna radiation pattern and 2. The height of the radar. For the first factor, we analyzed the radar radiation pattern integrated with our in-package lens in detail in Section 2.1, showing the antenna’s main beam occurs at a relative azimuth angle of θ = 0°. Figure 5 depicts the proper radar setup (radar position) at the height of “h” showing the main beam illuminates the subject’s torso. To obtain the best value for “h”, we referred to some facts about the average value of human height, lower body length and torso length. The average human height is about 164.5 cm, with a mean lower body length of about 98 cm and a mean torso length of about 48 cm [32]. Hence, we placed the radar at a height of 120 cm that will illuminate the walking person’s torso with its main beam (this height was confirmed in this work and our previous studies [11,21,25] with participants with different height, ranging from 150 cm to 210 cm). With this measurement setup, the chance to pick up strong returned signals from knee and arm motions is very low, and then the torso line can be selected from the occupied range bins by isolating the maximum signal. Unlike many studies [7,8,9,33,34,35], we suggest that both antenna patterns and signal processing techniques be considered in radar-based gait monitoring systems.

### 3.1. Experimental Results in a Hallway Environment

To show the effectiveness of our proposed radar system paired with our gait extraction algorithm, we evaluated system performance in a hallway filled with metal cabinets, shown in Figure 6a. This is a challenging scenario as the metal cabinets cause a strong reflection that causes significant multipath effects. We also considered this setup to be one of the “worst case” scenarios for where gait would plausibly be monitored (i.e., a hallway of a hospital). Figure 6b shows the front view of the fabricated system. Gait cycle parameters were extracted by asking four volunteers to walk back and forth three times by following marked steps (420 cm with six steps, each step is 70 cm for each lap). 

Note that for all tests in this work, reference values were extracted using a stopwatch and asking volunteers to follow a traced line that had marks the volunteers were asked to step on. To demonstrate radar sensor performance in hallway gait monitoring, we first examined radar results without the lens. Our proposed method (Figure 4) was then applied to the received signals of walking cycles to extract gait spatiotemporal parameters.

As shown in Figure 7, the total number of steps of the whole walking process is N = 57 without using the in-package lens, while the actual step count was 36. Moreover, the average walking speed and step length are 2.1 m/s and 90.3 cm, respectively, whereas the true value obtained by the stopwatch is 0.867 m/s. As seen, the extracted gait parameters are not accurate using the radar without the lens, while accuracy is one of the key factors in gait assessments [23]. As shown in Figure 3, the reason for these inaccurate results is due to the multipath effects and unwanted reflections from the walls.

Gait parameters were extracted using the system integrated with the in-package lens in the hallway. Based on the gait extraction algorithm proposed in Figure 4, the average walking speed of *V* = 0.83 m/s was obtained while the reference value was 0.87 m/s. To obtain other gait parameters, applying peak detection to the absolute values of the velocity of the torso, the step time of each cycle, step count, and the number of total steps were obtained, as shown in Figure 8. The total calculated number of steps is N = 36, which is exactly equal to the actual value. As shown, the number of steps and, thus, cadence is obtained with 100% accuracy. Note that the time to change direction (i.e., to turn at each end of the walk) was included in the gait speed calculation, but it was excluded in other gait parameters extraction. Furthermore, since the time of each step is known from Figure 8, step points are obtained from the range of the subjects. Then, each step length was calculated by subtracting two successive step points. Therefore, spatiotemporal gait parameters at each cycle, such as step points, step time, speed of torso, and step length were extracted. It should be mentioned that we provided gait instance parameters obtained from one participant to show the details of our proposed system. However, the average values of extracted parameters of the same participant are provided in Table 2. 

Additionally, the average error of extracted parameters of the four participants is provided in Table 2. In order to compare the gait values obtained in this paper with previous works, we listed a number of references in Table 3 with the reported error range, extracted gait parameters, the type of radar and the number of radars used for their experiment. As shown, according to [36], the accuracy obtained from our works is sufficient to detect meaningful changes over time. In [36], changes of 0.05 m/s have been deemed clinically meaningful. As shown, the error reported in our work is very low compared with other reported works, which is clinically meaningful. Moreover, we used only one FMCW radar that provided both spatiotemporal gait parameters at each gait cycle, while other works either used two radars to provide some detailed gait parameters or added an extra device such as a treadmill to provide these parameters. Therefore, while future work with a greater range of test conditions and more participants must be done to ascertain the accuracy of our algorithm, these preliminary results demonstrate the promising potential of our algorithm to monitor several aspects of gait in hallways accurately.

### 3.2. Experimental Results in a Clutter-Free Area

In this subsection, we provide results of gait tests performed in a clutter-free environemnt. There are two reasons for gait extraction in a clutter-free environment in this paper. Firstly, we compare the performance of our proposed gait extraction algorithm using the AWR1443Boost radar without the lens in a clutter-free environment, as shown in Figure 9, with the parameters extracted in the hallway using the radar integrated with the lens. We demonstrate that using the radar paired with the lens, gait parameters could be extracted in a cluttered environment such as a hallway as accurately as in a clutter-free environment. This demonstrates the effectiveness of the dielectric lens in mitigating multipath effects leading to an accurate gait extraction method. Secondly, with the defined setups in this subsection, as shown in Figure 9, the effects of the direction of walking on the performance of our proposed algorithm are shown. Similar to the hallway setup shown in Figure 6a, we asked volunteers to walk back and forth three times across the defined direction. 

A summary of the extracted values of walking at different angles is provided in Table 4. Table 4 shows the average extracted gait parameters in a clutter-free environment. As seen, our proposed gait extraction algorithm is very accurate in calculating the walking speed which is independent of the direction of walking. This is because walking speed is obtained based on the change in the position of the subject over time, not the micro-Doppler pattern. For other gait values, although the algorithm uses the Doppler patterns to find the step time, the accuracy is not significantly impacted by the direction of walking. The reason is that our proposed algorithm is based on the variation of the torso’s velocity during the stance and swing phases (it speeds up and slows down, creating a sawtooth shape) but not the actual value of the Doppler. In fact, as long as the walking cycle is the only influential variable on the radial velocity of the torso, our proposed algorithm extracts gait values accurately. In our experimental setup, since the relative angle between the radar and the subject was constant, the direction of walking did not impact the extracted results significantly. However, if the subject walks at different random angles in front of the radar, the angle’s effects on the micro-Doppler should be compensated to obtain reliable results. In this case, the azimuth information of the subject should be obtained using the MIMO feature of the radar, which is out of the scope of this paper as our focus is on hallway gait monitoring. It is worth mentioning that we realized that the worst case occurred when the subject walked perpendicular to the path of the radar illumination (90°). Although walking speed was extracted accurately (1.02 m/s while the reference value was 0.98 m/s), the algorithm did not perform very well in extracting other gait values; micro-Dopplers could not be obtained precisely at 90°. Comparing Table 2 and Table 4, it is clear that the in-package lens effectively removes and mitigates reflections from the walls such that walking parameters extracted in a hallway are almost the same as in a clutter-free environment. Therefore, our proposed system not only could be used for gait assessment in a hallway at long-term care facilities and hospitals but also at an individual’s home to prepare a day-to-day and frequent gait assessment. To the best of the authors’ knowledge, this is the first time that spatiotemporal gait values at each walking cycle are extracted using only one modified FMCW radar with a focused beam.

It should be pointed out that the radar we have been using is a low-power system. Its transmitted signals have less than 10 dBm of power. In comparison, Wifi is more than 20 dBm, and cellphones are in the order of 30 dBm. Power-wise this system is much lower in power than other wireless devices; thus, it could be used for the long term. Moreover, it uses high frequencies, which are harder to penetrate the human body. Electromagnetic absorption at these frequencies is more than 10 times less than those used for cellular bands and Wifi.

## 4. Conclusions

In this work, we addressed the challenges of gait assessment in a cluttered environment, such as a hallway lined with metal cabinets. To overcome the multipath effects, we proposed an easy and low-cost method of radar antenna modification to focus the field of view of a commercially available radar board. Our results suggest that using our add-on hyperbola-based dielectric lens antenna and implementing our gait extraction algorithm, a stand-alone gait assessment system could be developed using any FMCW radar. It should be emphasized that this work was designed for single subject tracking and monitoring. Our in-package lens antenna design should support relatively easy implementation with commercially available radars. Since the system is portable, easy-to-use, and low-cost, it could be installed in a variety of living environments, including long-term care, hospitals, or individuals’ homes, for the day-to-day gait of natural gait assessment in a way that requires no effort from the people being monitored.

## Figures and Tables

**Figure 1 sensors-23-00071-f001:**
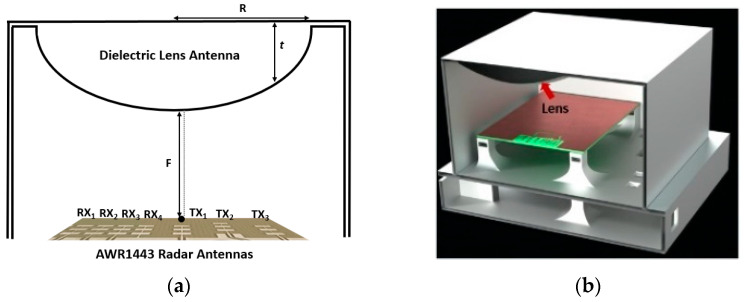
(**a**) Schematic of the in-package dielectric hyperbola-based lens antenna integrated with the AWR1443Boost radar, (**b**) Cross-view of 3D model of the designed system in SolidWorks with the radar/lens cover showing the encapsulated lens. TXs: transmitters, RXs: receivers.

**Figure 2 sensors-23-00071-f002:**
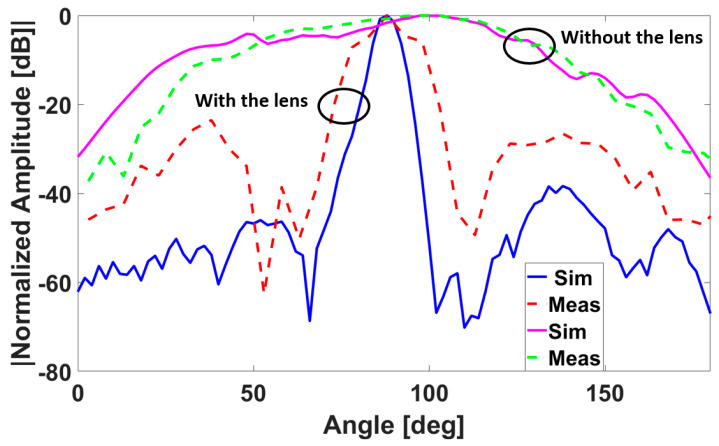
Simulated and measured patterns of the radar received power by the radar receiver Rx1 transmitted by Tx1 integrated with and without the lens.

**Figure 3 sensors-23-00071-f003:**
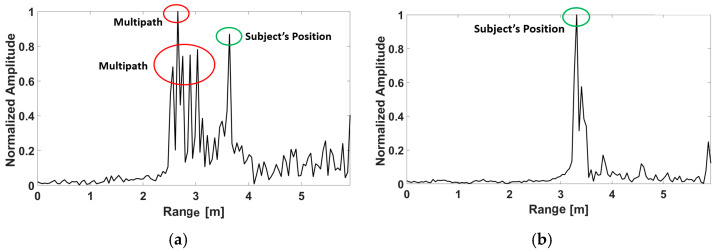
Range of a walking subject at frame 150 (**a**) without the lens (**b**) with the lens.

**Figure 4 sensors-23-00071-f004:**
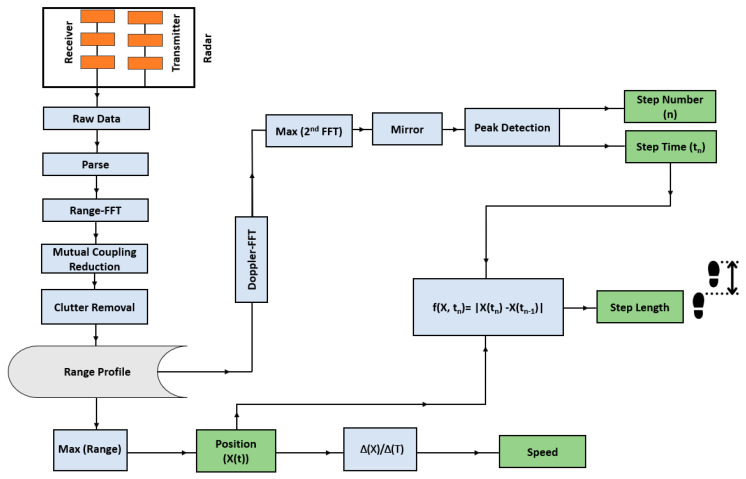
Proposed gait extraction algorithm (adapted from [11]).

**Figure 5 sensors-23-00071-f005:**
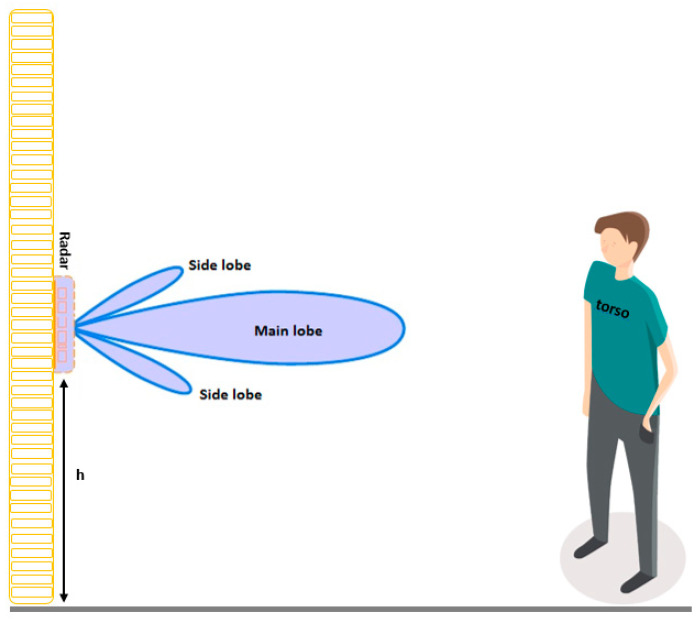
Proper radar position for gait monitoring using a single radar sensor (the main beam illuminates the walking person’s torso).

**Figure 6 sensors-23-00071-f006:**
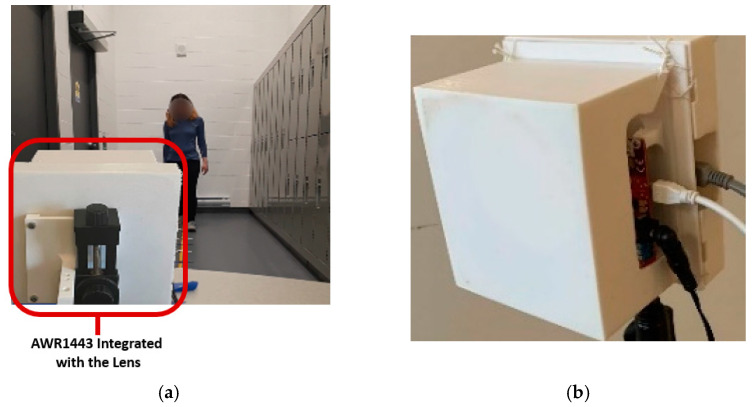
(**a**) experimental setup for hallway gait assessment (**b**) front view of the fabricated system.

**Figure 7 sensors-23-00071-f007:**
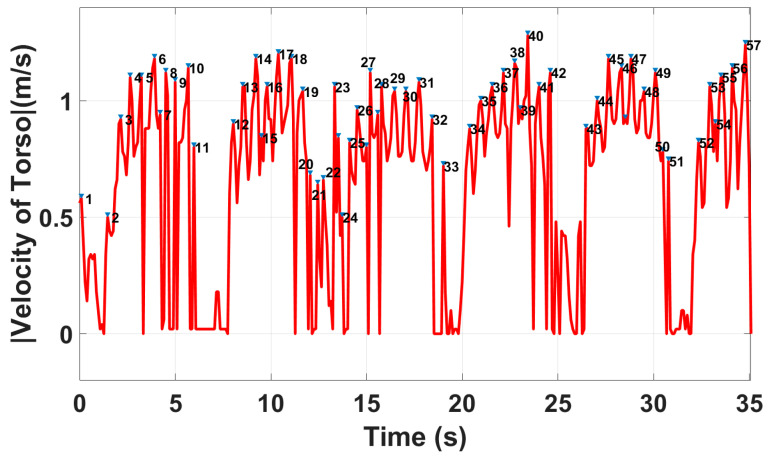
Peak detection algorithm applied to the absolute value of the velocity of the torso using the radar sensor without the lens.

**Figure 8 sensors-23-00071-f008:**
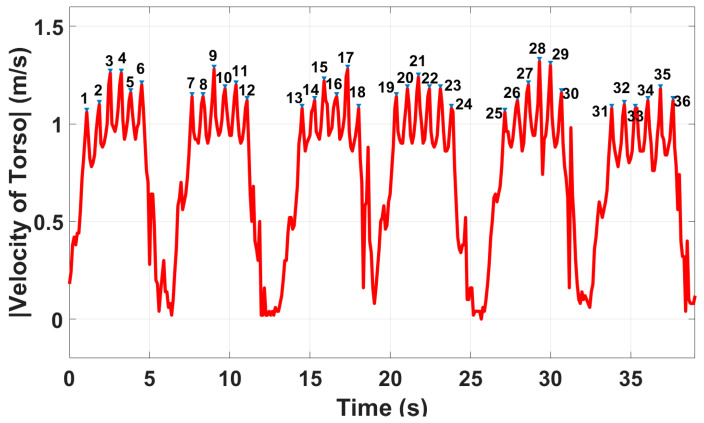
Peak detection algorithm applied to the absolute value of the velocity of the torso using the radar sensor integrated with the in-package lens.

**Figure 9 sensors-23-00071-f009:**
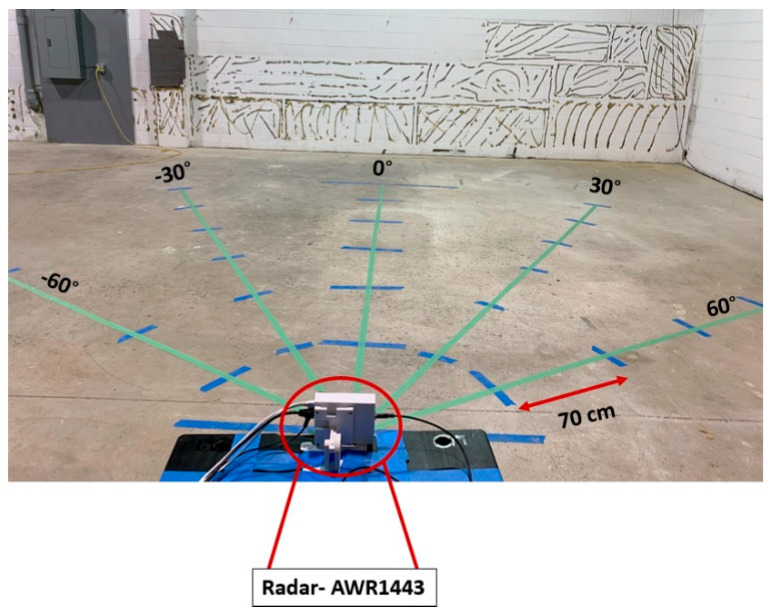
Experimental setup for gait assessment in a large space environment detection algorithm.

**Table 1 sensors-23-00071-t001:** Radar parameters, definition and values used for radar configuration to be used for the tests.

Characteristic	Characteristic Description	Specification
Start Frequency	The frequency of the radar signal will start at	77 GHz
Frequency Slope	The slope at which the frequency of the radar is increasing.	60 MHz/μs
Idle Time	The time between the previous chirp finishing and the frequency ramp starting	250 μs
Transmit Start Time	The time within the chirp where the transmitter is turned on	98 Μs
ADC Start Time	The time when the ADC starts sampling	10 μs
ADC Samples	The number of samples the ADC takes	64
ADC Sample Rate	The rate at which the ADC takes samples	2200 Ksps
Ramp End Time	The time when the frequency ramps finished	60 μs
Chirps/frame	The number of chirps per frame	256
Bandwidth	The difference between the maximum and the minimum frequency	3600 MHz

**Table 2 sensors-23-00071-t002:** Extracted gait values in a hallway.

	Speed (m/s)	Step Count	Step Length (cm)
Reference Value	0.87	36	70.0
Radar W/O the lens	2.10	57	90.3
Radar W/the lens	0.83	36	68.6
Average error for four participants (W/the lens)	0.013	+1.25	−2.33

**Table 3 sensors-23-00071-t003:** Comparison of the gait results obtained in this paper with other previous works.

Reference	Reported Error for Speed	Number of Radars	Type of Environment	Extracted Parameters	Radar Type and Other Required Devices
[37]	Not reported	1	Low clutter	mean walking speed, maximum leg velocity, maximum leg velocity, mean leg velocity in swing and stance phase, degree of variation of leg velocity in swing and stance phase,	Micro-Doppler
[38]	For 1.1. m/s walk (foot velocity error): 0.06 m/s to 0.17 m/s	2	Low clutter	Stride time, stance time, flight time, step time, cadence, stride length, step length, maximal foot velocity, maximal ankle velocity, maximal knee velocity, time instant of maximal knee velocity:	Continuous waves and treadmill
[39]	0.144 m/s	2	Low clutter	Foot velocity, torso velocity, step time	pulse-Doppler
[40]	For 10 GHz: slow walk: 0.4 m/s and normal walk: 0.14 m/s For 24 GHz: 0.5 m/s and normal walk: 0.06 m/s	1	Low clutter	Walking speed	10 GHz pulse-Doppler 24 GHz FMCW
[11]	0.0040 m/s to 0.043 m/s	1	High clutter	At each gait cycle: walking speed, maximum velocity of the torso, step length, number of steps, step points, step time, step count	FMCW radar
This work	0.0038 m/s to 0.045 m/s	1	High clutter	At each gait cycle: walking speed, maximum velocity of the torso, step length, number of steps, step points, step time, step count	FMCW radar paired with a hyperbolic lens

**Table 4 sensors-23-00071-t004:** Extracted gait values in a clutter-free at various angles.

Direction of Walking	Step Count	Step Length (cm)	Speed (m/s) Radar	Speed (m/s) Stopwatch
−60°	35	65.80	0.96	0.91
−30°	36	68.01	0.86	0.90
0°	36	69.10	0.95	0.96
30°	36	68.57	0.94	0.98
60°	33	65.06	0.95	0.96

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
