# Peer review of "Hallway Gait Monitoring System Using an In-Package Integrated Dielectric Lens Paired with a mm-Wave Radar"

_sensors, 2022, doi:10.3390/s23010071_

Round 1
Reviewer 1 Report
The paper proposes an mm-Wave based gait monitoring system that is resistant to multipath effects by integrating the radar with dielectric lens. The real experiments are conducted in real-life scenarios to show the effectiveness of the proposed system. However, the paper has several points that need attention, in terms of system design and performance evaluation.
1. For the system design, the major concerns are as follows:
1a. Novelty may be limited. The modified radar with integrated lens has been commonly used in gain improvement and directivity maximization. FMCW radar has also been widely used in gait recognition applications. It would be better if authors can point out the uniqueness of the work more explicitly.
1b. The benefit of adopting millimeter-wave over other sensing technologies is not clear. There are many WiFi-based gait monitoring systems, for instance, the system in [1], that can recognize the gait parameters from the estimated speed, which is even cheaper, while mmWave-based techniques require additional hardware (i.e., mmWave antennas and decoding chips). Authors may need to provide a more comprehensive comparison.
2. The system evaluation is not sufficient. The major concerns are as follows:
2a. Lack of details on setup. For example, what is the height of the mmWave radar? and what is the distance between the radar and the testing area?
2b. Lack of details on evaluation. For example, how many antennas are used to generate the results in table 1 and table 2? Will the performance be improved if more antennas are used?
2c. Some other information is missing. For example, what is the maximum sensing range for the modified mmWave radar? Also, It would be interesting to investigate how estimation accuracy is affected by the distance between the user and the testing area.
2d. The motivation for experiments in ‘Section 3.2 - the experimental results in a large area.’ is unclear. The authors mentioned ‘the purpose of this subsubsection is to compare the performance of … without the lens in a large environment, … with the parameters extracted in the hallway using the radar integrated with the lens’, which suggests the experiments in a large area are conducted with radar without lens integrated. Even if the results show high accuracy in estimating gait parameters, the experiments seem to be unnecessary as the focus of the work is the usage of lens. To investigate the impact of walking direction and prove the generalization of the system from hallways to large areas, the experiments should be done with modified radar with lens.
2e. The authors’ previous work [2] is mentioned multiple times. The two works tend to solve the same problem of multipath effects from different perspectives. The previous one is based on the postprocessing of signals, while the current one directly relies on the extra lens. Apart from the principles, a more detailed comparison between these two methods should be provided and tabulated, including the exact speed estimation error, range resolution, and quantitative complexity.
3. Other concerns:
3a. Authors may need to mention more related mm-Wave-based works.
3b. The format of reference [8] is not official. It is shown explicitly as ‘under review’.
[1] C. Wu, F. Zhang, Y. Hu, and K. J. R. Liu, "GaitWay: Monitoring and Recognizing Gait Speed Through the Walls," in IEEE Transactions on Mobile Computing, vol. 20, no. 6, pp. 2186-2199, 1 June 2021, doi: 10.1109/TMC.2020.2975158.
[2] H. Abedi, J. Boger, P. P. Morita, A. Wong and G. Shaker, "Hallway Gait Monitoring Using Novel Radar Signal Processing and Unsupervised Learning," in IEEE Sensors Journal, vol. 22, no. 15, pp. 15133-15145, 1 Aug.1, 2022, doi: 10.1109/JSEN.2022.3184188.
Author Response
Dear Reviewer,
Thank you for your insightful comments on our paper. We have highlighted the changes within the manuscript. Here is a point-by-point response to your and the reviewers’ comments and concerns.
Reviewer: 1
Comments and Suggestions for Authors
The paper proposes an mm-Wave based gait monitoring system that is resistant to multipath effects by integrating the radar with dielectric lens. The real experiments are conducted in real-life scenarios to show the effectiveness of the proposed system. However, the paper has several points that need attention, in terms of system design and performance evaluation.
- For the system design, the major concerns are as follows:
Comment 1:
1a. novelty may be limited. The modified radar with integrated lens has been commonly used in gain improvement and directivity maximization. FMCW radar has also been widely used in gait recognition applications. It would be better if authors can point out the uniqueness of the work more explicitly.
Response 1:
Based on the provided comments, we realized that the reviewer sees this manuscript either purely from an antenna design perspective or the type of radar. In contrast, this paper presents a unique method to solve a real-life problem (multipath effects) in hallway gait monitoring. We would like to request the reviewer to look at this manuscript on how to use and modify a well-formulated material to solve an issue in a real-life application.
We agree that a dielectric lens concept by itself does not represent a novel contribution. As the reviewer mentioned, extensive research has been done in this field. Where we see the value of the work is the whole implementation, including the choice of the package-friendly lens and inclusion as part of the package design to be paired with a commercially available radar. We achieved a reliable hallway gait assessment system using a commercially available radar without altering the radar PCB board and while enabling a simple flat packaging for the whole system. Our work provided a useful and valuable approach for many radar-based applications that do not desire the use of hemispherical lens designs.
In more detail, we used the lens design for multipath mitigation, a novel and unique method in gait monitoring systems, to solve the multipath issues we encountered using the commercially available radar. As stated in the manuscript, almost all the published gait monitoring campaigns were conducted in a clutter-free area where there are fewer or any multipath reflections. However, the ultimate goal in gait analysis using radar sensors is to have an easy-to-implement and portable device for day-to-day gait monitoring at individuals’ homes, long-term care facilities or hospitals where there are lots of objects around, creating multipath reflections. In our previous work, Ref [11], we showed how multipath reflections in a cluttered environment could impact the accuracy of gait parameters using the common method in gait analysis using radar sensors. Mainly, there are some sophisticated algorithms to remove or mitigate multipath reflections that are computationally costly. These sophisticated algorithms need a powerful system. However, we proposed a lens solution that removes multipath reflections without implementing any sophisticated algorithms. Therefore, our proposed system is inexpensive and portable, enabling the common gait monitoring algorithm to be implemented in a radar DSP or a Raspberry Pi without the need for an extra desktop or laptop system. Consequently, we believe that what we presented in this paper provides an innovative method for gait extraction in narrow hallways such as those typically encountered in hospitals and long-term care homes. In the last paragraph of the Introduction on page 3, we explained our contributions in detail.
Regarding the FMCW radar, we do not claim that this is the first time such a radar has been used for gait extraction. As mentioned above, our novelty is not using FMCW radar in this paper, although this type of radar helps us extract gait parameters easily at each single gait cycle.
Comment 2:
1b. The benefit of adopting millimeter-wave over other sensing technologies is not clear. There are many WiFi-based gait monitoring systems, for instance, the system in [1], that can recognize the gait parameters from the estimated speed, which is even cheaper, while mmWave-based techniques require additional hardware (i.e., mmWave antennas and decoding chips). Authors may need to provide a more comprehensive comparison.
Response 2:
Firstly, we should distinguish gait recognition and gait monitoring or gait extraction. For the first one, the purpose is to identify humans based on their gait patterns which has been extensively done using WiFi and radar systems. For the second one, radar systems have been mainly used. There are various reasons why WiFi is less explored for gait extraction. We explain some parts briefly here. The main purpose for gait extraction is because gait values are considered health indicators and predictors for many diseases. For example, cognitive functions decline 12 years before mild cognitive impairment. One of the key factors to make gait values applicable is accuracy. According to Ref [36] in our manuscript, to be clinically meaningful, the speed error should be less than 0.05 m/s. However, from WiFi, this accuracy is not easily achievable. For example, GAITWAY, Ref [1] achieved a median error of 0.12 m/s in gait speed monitoring, while this accuracy would not be clinically acceptable as a health predictor/indicator. Not to mention that many advanced signal processing methods are needed for WiFi-based methods, such as what was proposed in Ref [1]. Regarding mm-wave, there are various advantages as follows:
- The high attenuation in mm-wave frequencies provides high isolation between the co-located operating radars, even if they are separated by a few meters.
- Tiny displacements in mm-wave are comparable to the wavelength; thus, they can be detected.
- Working at higher bandwidth, there would be less interference with other electromagnetic-based devices, especially if this device is used at hospitals or LTCs.
- Low-power system. Its transmitted signals have less than 10 dBm of power. In comparison, WiFi is more than 20 dBm. Power-wise this mm-wave system tends to be much lower in power than other wireless devices; thus, it could be used for the long term.
- Using high frequencies, it is harder to penetrate the human body. Electromagnetic absorption at these frequencies is more than 10 times less than those used for cellular bands and WiFi.
Comparing radar-based sensors Vs WiFi-based sensors is an interesting topic that requires another publication which is out of the scope of this paper.
Because of the aforementioned reasons, although we added the reviewer’s suggested paper in our citations Ref [1], we believe that talking more about WiFi-based gait recognition and extraction method in our paper would confuse readers and deviate them from realizing the values of our work which is system implementation for gait analysis in a high-cluttered environment.
- The system evaluation is not sufficient. The major concerns are as follows:
Comment 3:
2a. Lack of details on setup. For example, what is the height of the mmWave radar? and what is the distance between the radar and the testing area?
Response 3:
To address this comment, we provided detailed information about the radar setup, how we achieved this setup, radar configuration (in Table 1) and etc., on pages 7 and 8. We added Figure 5 since we conducted this study in our works, but we have not seen any publications considering the proper installation of the radar sensor. We believe this section would be interesting and helpful for many readers in this field.
Comment 4:
2b. Lack of details on evaluation. For example, how many antennas are used to generate the results in table 1 and table 2? Will the performance be improved if more antennas are used?
Response 4:
As mentioned on page 7, based on our proposed algorithm, only one transmitter and one receiver is enough (this is one of the advantages of using the lens, as mentioned in the Introduction section as well). So we used only one transmitter and one receiver in this paper. It should be noted that since our main focus in this paper is on showing the performance of our algorithm integrated with the in-package lens, the gait parameters of a single subject were monitored. Therefore, only one transmitter and one receiver are sufficient. However, as mentioned on page 7, the reason for designing the in-package lens for the radar with multiple transmitters and receivers is to use the same proposed system to have a reliable comparison with our previous work and for multiple people in our future work. In terms of reliable comparison, as you know, there are lots of factors affecting radar performance, such as range resolution, idle time, maximum achievable range and etc. Finding two radars that could provide identical parameters is not easy, so we used the same radar that could provide a reliable comparison.
Regarding adding more antennas, even using the MIMO channels, as shown in Ref [11] and explained in the Introduction section on page 2, paragraph 5, as well as page 7, we still require an algorithm to mitigate the multipath effects to obtain accurate gait parameters. Again, we would like to emphasize that our main challenge is multipath effects that need either sophisticated signal processing methods or a thin antenna radiation pattern, which we discussed in detail.
Comment 5:
2c. Some other information is missing. For example, what is the maximum sensing range for the modified mmWave radar? Also, It would be interesting to investigate how estimation accuracy is affected by the distance between the user and the testing area.
Response 5:
We believe this comparison goes beyond the scope of this paper. Our focus is on using the lens to help with multipath issues rather than improving the range. Note that if we increase the range, we will not get the same range resolution or maximum achievable velocity. There are trade-offs in these parameters. As seen in our previous work Ref [11], if you check the range resolution, we do not have the identical resolution as we do here in this work. So, it is kind of impossible to provide a fair comparison unless we just care about range maximization without caring about the resolution limits, which is not the purpose of this work.
Comment 6:
2d. The motivation for experiments in ‘Section 3.2 - the experimental results in a large area.’ is unclear. The authors mentioned ‘the purpose of this subsubsection is to compare the performance of … without the lens in a large environment, … with the parameters extracted in the hallway using the radar integrated with the lens’, which suggests the experiments in a large area are conducted with radar without lens integrated. Even if the results show high accuracy in estimating gait parameters, the experiments seem to be unnecessary as the focus of the work is the usage of lens. To investigate the impact of walking direction and prove the generalization of the system from hallways to large areas, the experiments should be done with modified radar with lens.
Response 6:
As explained on page 11, section 3.2, the reason for this subsection was not to show the lens performance. The reason was to show the performance of our proposed algorithm from different angles. We believe this section provides some good examples of the effect of the angle on our signal processing method for readers. Moreover, this subsection shows the results obtained from our lens system are almost the same as what we have achieved in a clutter-free environment.
Regarding the suggestion about using the lens at different angles, since the radiation patterns of our lens are sharp, the lens received signals at other angles (for example, 30o) would be very low. Actually, this is the main point of the lens design in this paper which was to sharpen the beam and remove reflections/ multipath effects from other angles). Due to the lens’s high directivity, signals at other angles would be significantly weakened, so we do not have good detection at other angles. We designed this package-friendly lens antenna (with a flat package-cover) for the hallway environment where people walk in a straight line, as mentioned on page 2, paragraph 4.
Comment 7:
2e. The authors’ previous work [2] is mentioned multiple times. The two works tend to solve the same problem of multipath effects from different perspectives. The previous one is based on the postprocessing of signals, while the current one directly relies on the extra lens. Apart from the principles, a more detailed comparison between these two methods should be provided and tabulated, including the exact speed estimation error, range resolution, and quantitative complexity.
Response 7:
To address this comment, in Table 3, we provided the average error obtained in our previous work and explained the differences in detail on pages 10 and11.
- Other concerns:
Comment 8:
3a. Authors may need to mention more related mm-Wave-based works.
Response 8:
We added more related works, as requested!
Comment 9:
3b. The format of reference [8] is not official. It is shown explicitly as ‘under review’.
Response 9:
It was corrected in this version!

Reviewer 2 Report
This work is interesting to go through. I would like to suggest the authors in order to consider the following points to be considered.
1. More description of Equation 1 is needed.
2. Both Table 1 and Table 2 must be explained in more detail.
3. Please place the Figures and Tables in the proper places as you describe them.
-----------------------------------------
1. What is the main question addressed by the research?
This work mainly tries to address day-to-day long-term monitoring of gait parameters (e.g., speed, step points, step time, step length, and step count) that may be the signs of changes in mobility, cognition, and frailty of older adults. Basically, the authors tried to overcome the multipath effects in gait assessment by proposing a method of radar antenna modification. The authors claim that they achieve it by designing an in-package hyperbola-based lens antenna integrated with a radar module package.
2. Do you consider the topic original or relevant in the field? Does it
address a specific gap in the field?
This work is mainly based on the authors' previous work [5] and [11]. It requires showing how this work is significantly different from the authors' previous works rather than simply a combination work of both ones. According to the authors, they have proposed two solutions: a new signal processing and radar antenna modification and it would have been better if they have implemented their approach in sensor technology like a Raspberry Pi [8]. Therefore, this work is more inclined to signal processing in the communication domain than pure sensor technology.
3. What does it add to the subject area compared with other published
material?
There is knowledge addition to the authors’ previous work. The authors adapted the gait extraction algorithm from their previous work [5] with its implicit demerits of requiring a powerful signal processing methodology.
4. What specific improvements should the authors consider regarding the
methodology? What further controls should be considered?
In a real scenario, older adults fall down in arbitrary places like the bathroom. Monitoring them in such a place is more critical (than in the hallway and corridor) to know the exact reason for the imbalance and eventual falling down of the person. It is needed to mention why the authors would like to monitor older adults? Problem statement could be clearly mentioned in this paper. The quality of the paper can be further improved with the inclusion of comparative results with other similar work.
5. Are the conclusions consistent with the evidence and arguments presented
and do they address the main question posed?
I would like to suggest to rewrite the conclusion part considering a clear scope of the work with trade-offs of the proposed method and future work.
6. Are the references appropriate?
References are appropriate but self-citations.
7. Please include any additional comments on the tables and figures
Regarding figures, authors are suggested to synchronize the placement of the figures and their descriptions. Figure 4 is mentioned in the paper after talking figure 5 and there is no reasonable level of description of the figure 4. It is unusual to see the placement of table 1, figure 8, and table 2 after the conclusion.
Author Response
Dear Reviewer,
Thank you for your insightful comments on our paper. We have highlighted the changes within the manuscript. Here is a point-by-point response to your and the reviewers’ comments and concerns.
Reviewer: 2
Comments and Suggestions for Authors
This work is interesting to go through. I would like to suggest the authors in order to consider the following points to be considered.
Comment 1:
- More description of Equation 1 is needed.
Response 1:
We added two paragraphs on pages 3 and 4 to explain this equation in detail.
Comment 2:
- Both Table 1 and Table 2 must be explained in more detail.
Response 2:
We explained them in detail in this version.
Comment 3:
- Please place the Figures and Tables in the proper places as you describe them.
Response 3:
All plots are placed on their right pages!
Comment 4:
- What is the main question addressed by the research?
This work mainly tries to address day-to-day long-term monitoring of gait parameters (e.g., speed, step points, step time, step length, and step count) that may be the signs of changes in mobility, cognition, and frailty of older adults. Basically, the authors tried to overcome the multipath effects in gait assessment by proposing a method of radar antenna modification. The authors claim that they achieve it by designing an in-package hyperbola-based lens antenna integrated with a radar module package.
Response 4:
Thank you for pointing this out!
Comment 5:
- Do you consider the topic original or relevant in the field? Does it
address a specific gap in the field?
This work is mainly based on the authors’ previous work [5] and [11]. It requires showing how this work is significantly different from the authors’ previous works rather than simply a combination work of both ones. According to the authors, they have proposed two solutions: a new signal processing and radar antenna modification and it would have been better if they have implemented their approach in sensor technology like a Raspberry Pi [8]. Therefore, this work is more inclined to signal processing in the communication domain than pure sensor technology.
Response 5:
To address this comment, in Table 3, we provided the average error obtained in our previous work and explained the differences in detail on pages 10 and 11. As shown, both systems are quite accurate in extracting gait parameters, but the main difference is the computational cost that Ref [11] has.
As described on pages 5 and 6 in detail, although both works address the same issue (multipath effects), we pointed out two different approaches. Each method has its advantages and disadvantages. As mentioned, our previous work needed more sophisticated algorithms, whereas this work overcame the multipath effects by hardware modification. We detailed the difference in this paper and strongly believe these two papers will be very helpful for many researchers in this field, not only for gait analysis but also for other radar-based applications. Our works show how to solve an issue in a real-life application by different methods. We showed how to implement radar signal processing in a Raspberry Pi in our previous work, which is under review in the IEEE IoT journal, so we believe talking about this process would conflict with our other work. Actually, we put several radars integrated with a Raspberry Pi in a local long-term care facility Ref [27]. However, we believe covering these details is out of the scope of this paper.
Comment 6:
- What does it add to the subject area compared with other published
material?
There is knowledge addition to the authors’ previous work. The authors adapted the gait extraction algorithm from their previous work [5] with its implicit demerits of requiring a powerful signal processing methodology.
Response 6:
Thank you for pointing this out!
Comment 7:
- What specific improvements should the authors consider regarding the
methodology? What further controls should be considered?
In a real scenario, older adults fall down in arbitrary places like the bathroom. Monitoring them in such a place is more critical (than in the hallway and corridor) to know the exact reason for the imbalance and eventual falling down of the person. It is needed to mention why the authors would like to monitor older adults? Problem statement could be clearly mentioned in this paper. The quality of the paper can be further improved with the inclusion of comparative results with other similar work.
Response 7:
To address this comment, we provided several pieces of information about gait, the reason for gait analysis, and also fall in the Introduction section on pages 1 and 2 (paragraphs 1, 2 and 3). Regarding hallway/corridor gait monitoring, as explicitly mentioned on page 2, paragraph 4, a challenge in gait assessment is that human walking contains the micro-Doppler signature that is dependent on the direction of the motion. To overcome the dependency on the relative angle between the radar and a walking person, we decided to monitor human gait in a corridor or hallway, as this is something that is in virtually every place people live and results in people performing a relatively straight line of walking in a natural way several times a day. We agree that older adults fall down in the bathroom or bedroom, but we do not cover falls in this paper. The analysis of gait is more proactive or preventive, which could alert health-wise issues a person might have. We should mention that since accuracy is more important, walking tests in a hallway would be more accurate as people walking in a straight line.
Regarding the comparison, we compared the work presented in this paper with our previous works Ref [11] and other references in Table 3. The corresponding narration is provided on pages 10 and 11.
Comment 8:
- Are the conclusions consistent with the evidence and arguments presented
and do they address the main question posed?
I would like to suggest to rewrite the conclusion part considering a clear scope of the work with trade-offs of the proposed method and future work.
Response 8:
We revised the conclusion section accordingly.
Comment 9:
- Are the references appropriate?
References are appropriate but self-citations.
Response 9:
We added more references in this version.
Comment 10:
- Please include any additional comments on the tables and figures
Regarding figures, authors are suggested to synchronize the placement of the figures and their descriptions. Figure 4 is mentioned in the paper after talking figure 5 and there is no reasonable level of description of the figure 4. It is unusual to see the placement of table 1, figure 8, and table 2 after the conclusion.
Response 10:
Figures are moved to be located right after they are mentioned in the paper.

Round 2
Reviewer 1 Report
The authors have addressed all my previous concerns.